# Electrochemical Deposition and Investigation of Poly-9,10-Phenanthrenequinone Layer

**DOI:** 10.3390/nano9050702

**Published:** 2019-05-06

**Authors:** Povilas Genys, Elif Aksun, Alla Tereshchenko, Aušra Valiūnienė, Almira Ramanaviciene, Arunas Ramanavicius

**Affiliations:** 1Department of Physical Chemistry, Faculty of Chemistry and Geosciences, Vilnius University, LT-03225 Vilnius, Lithuania; povilasg.tar@gmail.com (P.G.); elifaksunn@gmail.com (E.A.); alla_teresc@onu.edu.ua (A.T.); ausra.valiuniene@chf.vu.lt (A.V.); 2NanoTechnas–Center of Nanotechnology and Materials Science, Faculty of Chemistry and Geosciences, Vilnius University, LT-03225 Vilnius, Lithuania; almira.ramanaviciene@chf.vu.lt; 3Department of Chemistry, Faculty of Arts and Sciences, Dumlupınar University, TR-43100 Kütahya, Turkey; 4Department of Experimental Physics, Faculty of Mathematics, Physics and Information Technologies, Odesa National I.I. Mechnikov University, Pastera 42, 65023 Odesa, Ukraine; 5Laboratory of Nanotechnology, State Research Institute Centre for Physical Sciences and Technology, LT-10257 Vilnius, Lithuania

**Keywords:** electrochemical polymerization, poly-9,10-phenanthrenequinone, modified electrodes, conducting polymers, redox polymers, electrochemical impedance spectroscopy

## Abstract

In this research, a 9,10-phenanthrenequinone (PQ) was electrochemically polymerized on a graphite rod electrode using potential cycling. The electrode modified by poly-9,10-phenanthrenequinone (poly-PQ) was studied by means of cyclic voltammetry, electrochemical impedance spectroscopy, atomic force microscopy and scanning electron microscopy. The poly-PQ shows variations in growth pattern depending on the number of potential cycles for the initiation of polymerization. Formed poly-PQ layer demonstrates good electric conductivity, great degree of electrochemical capacitance and unique oxidation/reduction properties, which are suitable for broad technological applications, including applicability in biosensors, supercapacitors and in some other electrochemical systems.

## 1. Introduction

Polymer-modified electrodes are mainly applied in the manufacturing of batteries or supercapacitors [1], organic light-emitting diodes [2] and biosensors [3,4,5]. In one example, the main feature of such organic materials is their specific gas sensing surface phenomena [6] that is determined by the polysiloxane polymer and its interaction with the electrode and/or solution, in which they were used. Polymers deposited on a substrate surface can possess various properties including advanced electrical conductivity [1,7] or insulating properties [8], redox mediating capabilities [9], interesting topographic features [10], specific adhesive and/or binding properties [11], etc. The modification of electrically conducting surfaces (e.g., electrodes) with polymers possessing a particular function can be performed by many well-established methods, for example by dissolving the polymer in a solvent and casting of this solution on electrode surface [3], polymerization using various chemical initiators [12] or altered environment conditions [13], electrochemical polymerization directly onto the electrode surface [2,14,15]. Polymer-modified electrodes can be composed of more than one functional layer, e.g., they can be multilayered [16,17], or additionally modified with some functionalized layers that can be formed during or after formation of initial polymer layer [18], etc.

The application of electrochemically deposited polymers currently is rapidly expanding [13]: including their usage in actuators [1], gas separation membranes [16], sensors [13] and charge storage devices [1,13]. Electrochemically deposited polymers can be encountered in a range of charge storage devices such as electrolytic capacitors and high capacity rechargeable batteries [1,13]. The key property of the polymers used in such systems is the ability of the polymer to accumulate and to store electrical charge. For example, while the still widely used aluminum-based capacitor cells achieve the charge storage by the accumulation of charges onto the oxide coated aluminum foil surface [19], in contrast a conducting polymer, polypyrrole-based electrolytic capacitor is storing the charge by undergoing redox reactions in the polymer backbone and therefore such strategy enables to store significantly higher amount of charge per volume unit of the capacitor [1,2].

Polyazines are macromolecules, which are formed by the polymerization of phenanthrolines, phenothiazines, phenoxazines or phenazines. This class of the polymers has recently become popular in biosensorics due to their advanced charge-transfer properties [20]. It is reported that phenanthroline and its derivatives can be used for electrochemical detection of DNA hybridization [21] and electrocatalytic oxidations of nucleotides [22]. Moreover, phenothiazine-based polymers are interesting due to good adhesion to a carbon surface [23]. New phenothiazine derivative (bis-phenothiazin-3-yl methane) has been formed on glassy carbon (GC) electrode and has been applied for electrocatalytic oxidation of NADH [24]. Additionally, bioelectrocatalytical properties of phenoxazine mediators can be well applied in the development of biosensors because their carboxylate group can be exploited for covalent binding of enzymes [25]. 9,10-phenanthrenequinone possesses a quinonic structure and it has one phenanthrene group, which is responsible for very attractive redox properties of this molecule [26,27]. Such properties of 9,10-phenanthrenequinone makes it an interesting object for the electrochemical applications including usage in redox mediating systems of biosensors. Therefore, the PQ was reported as an electrode modifier suitable for facilitated oxygen reduction [28]. In addition, some attempts to apply PQ as a semiconducting layer were reported [29]. On one hand, the functionality of quinonic structure located in PQ-structure permits it to undergo reversible both one-electron and two-electron-based reductions that are generating a semiquinone radical or quinone, respectively [26]. On the other hand, the phenanthrene group—which is also involved into aromatic system of PQ—can interact with graphene, carbon nanotubes and other carbon-based materials via the *π-π* stacking with the *π-π* conjugated structure of PQ [30]. It should be noted that the simple but still highly ordered *π-π* conjugated structure also provides for the PQ molecule considerable thermal stability at elevated temperatures [26] and stability at ambient conditions [31].

The aim of this research was to electrochemically polymerize the monomeric compound–9,10-phenanthrenequinone and to evaluate electrochemical capacitance with some other electrochemical and structural properties of formed polymer layer.

## 2. Experimental Section

### 2.1. Chemicals

Acetate-phosphate buffer solution (A-PBS), pH 6.0, containing 50.0 mM of sodium acetate, 50 mM monopotassium phosphate and 50.0 mM of disodium phosphate with 100.0 mM of KCl was prepared. All chemicals were of analytical grade and purchased from AppliChem (Darmstadt, Germany), Sigma-Aldrich (Hamburg, Germany), Roth (Karsruhe, Germany), Scharlau (Barcelona, Spain) (in series for A-PBS buffer solution preparation), Sigma-Aldrich (Hamburg, Germany) (9,10-phenanthrenequinone), Alfa Aesar (Kandel, Germany) (tetra-*n*-buthylammonium perchlorate) companies unless it is noted.

### 2.2. Pre-Treatment of the Working Electrode

Electrode modification was carried by means of electrochemical polymerization of PQ. To avoid contamination by any accidental oxidation products and to obtain clean electrode surface, electrodes were boiled in a 4:1 mixture (by volume) of 25% ammonia (from Chempur (Piekary Slaskie, Poland)) and 30% H_2_O_2_ (from AppliChem (Darmstadt, Germany)) for about 10 min, washed with acetone, ethanol and distilled water prior to usage and dried at room temperature. The surface of the graphite rod (GR) electrode was polished with a fine emery paper of 3 roughnesses (firstly with grit number of P 300, then P 1000 and finally with P 2500) rotating on a workshop-made polishing machine. To achieve the best repeatability of results the working area of the graphite electrode was limited to only a circular tip of 0.071 cm^2^, which was additionally polished by paper to mirror smoothness to minimize the charge holding double electric layer disturbances and to form a smooth polymer layer on the surface of the electrode, although in later topography studies it can be seen that there are some imperfections invisible to the naked eye, but such deviations practically do not influence the results. The polymerization solution consisted of 0.001 M of PQ and 0.1 M of tetra-*n*-butylammonium perchlorate both dissolved in acetonitrile. Initial parameters, which were applied for PQ polymerization, were chosen taking into account the parameters which were applied previously in the polymerization of 1,10-phenanthroline-5,6-dione [32], which is one of many PQ derivatives. Electropolymerization of PQ was performed by potential cycling, which was followed by simultaneous registration of cyclic voltammograms. Applied potential interval was optimized in the range from +0.5 V to +2.5 V vs. Ag/AgCl/KCl_(3M KCl)_, at a scan rate of 0.1 V/s.

### 2.3. Electrochemical Measurements

Electrochemical measurements were performed using a potentiostat/galvanostat AUTOLAB PGSTAT 30 from Eco Chemie (Utrecht, The Netherlands), which was operated with the FRA and GPES software, provided by Eco Chemie (Utrecht, The Netherlands), in an electrochemical cell with A-PBS, pH 6.0, solution of different compositions depending on objective of investigations. A three-electrode configuration system inside of a Faraday-cage at ambient temperature (25 °C) was used for all electrochemical investigations, it was assembled from an Ag/AgCl in 3M KCl (Ag/AgCl/KCl_(3M KCl)_) Metrohm 6.0733. 100 reference electrode, a platinum wire auxiliary electrode and a graphite rod electrode, which was used as a working electrode, to which the reference electrode was kept as close as possible. To keep the studied system as simple as possible and to avoid any other reactions taking place no additional redox mediators like the ferri/ferro system were used. The electrochemical impedance spectra (EIS) were recorded in the frequency range from 40 kHz to 5 mHz at 0.0 V vs. Ag/AgCl/KCl_(3M KCl)_ working electrode potential while applying potential perturbation at 10 mV amplitude vs. Ag/AgCl/KCl_(3M KCl)_. For further data analysis the cell′s responses to applied electrochemical perturbation at different frequencies are presented as Nyquist plots. For extracting EIS-characteristics from the impedimetric data under selected equivalent circuits an experimental data fitting program ZView (v2.3) was used and standard error of all calculated equivalent circuit elements as it was indicated by the program was lower than 10% except that value of some Warburg elements. The simulated spectra are not presented to avoid a confusing overlapping with experimental points in most of the plotting area. Polymerization and measurements were carried out in a closed vessel to avoid solution evaporation or the influence of any atmospheric changes.

### 2.4. Imaging of the Electrode Surface by Atomic Force Microscope and Scanning Electron Microscope

Surface roughness was evaluated using atomic force microscope (AFM) Veeco Bioscope/Catalyst (Santa Barbara, CA, USA). The topography images were registered in air using contact mode with a soft silicon nitride tip covered with reflective gold coating. The spring constant of the tip was 0.06 N/m, scan rate was 0.5 Hz. All images were processed using the program NanoScope 8.0 provided by Veeco. Scanning electron microscope (SEM) from Hitachi TM 3000 (Tokyo, Japan) was also used in this study. All SEM scans were obtained at a 15 kV electron acceleration voltage at magnifications of 50×, 1000×, 10,000× and 30,000× with custom brightness and focus settings.

## 3. Results and Discussion

### 3.1. Investigation of Electrochemical Formation of Poly-PQ Layer

#### 3.1.1. Investigation of Polymerization by Potential Cycling

In this work we performed the electrochemical polymerization of PQ, because this method allows us to change and control some parameters, which are important for the efficiency of polymerization reaction [2,13,32,33]. Out of many others electrochemical polymerization methods we have selected the potential cycling-based method due to our previous experience in electrochemical polymerization of some related organic compounds. In addition, this method has some advantages over other electrochemical methods, because it is a simple and built-in as a standard electrochemical method “cyclic voltammetry” into most potentiostats, and it allows in situ evaluation of obtained data [34]. Before PQ polymerization procedure and after each CV cycle the electrochemical impedance spectra were recorded directly in the polymerization solution for more detailed evaluation of the polymerization process [35,36].

To determine the optimal interval of potential cycles for electrochemical poly-PQ synthesis on GR electrode, different potential intervals were tested from +1.2 V to +2.7 V; from +0.5 V to +4.0 V; from 0.0 V to +2.7 V; from −0.5 V to +2.0 V and from 0.5 V to 2.5 V. The polymerization (3 potentials scans) was performed in acetonitrile applying scan rate of 0.1 V/s vs. Ag/AgCl/KCl_(3M KCl)_. During the initial search for optimal values no significant oxidation and/or reduction peaks in cyclic voltammograms were observed cycling potential in the intervals from +1.2 V to +2.7 V, from +0.5 V to +4.0 V, and from −0.5 V to +2.0 V. The reduction peak about +1.0 V was observed cycling potential in the interval from 0.0 V to +2.7 V. The best results were obtained cycling potential in the interval from 0.5 V to 2.5 V. These conditions are already sufficient for the establishment of an observable reduction peak and monitoring the cycle′s changes in the experiments discussed below.

Modification of electrode with poly-PQ at selected optimal conditions (cycling potential in the interval from 0.5 V to 2.5 V applying scan rate of 0.1 V/s vs. Ag/AgCl/KCl_(3M KCl)_) was performed in further research. Cyclic voltammograms recorded during potential cycling are shown in Figure 1. All cycles are of almost similar shape with reductive peak at about 1.37 V, which is decreasing in magnitude and shifting to lower potential with every cycle. The explanation for this could be that the concentration of reducing species (9,10-phenanthrenequinone) is decreasing during this procedure and that the newly formed poly-PQ layer is lowering the energy levels required to initiate the PQ polymeric chain growth on the surface of graphite electrode, although the growing polymer layer creates additional resistance that has to be overcome by electrons during reduction of PQ monomer, this could explain the reduction peak′s decrease in current. After the integration of the first nine reduction peak areas and applying the amount of calculated charge into the equation of Faraday′s law of electrolysis, and assuming that monomer reduction process uses one electron per molecule, it was determined that the amount of reducing species formed in used volume of polymerization solution (0.634 mol), indeed, this is confirmed to be of the same magnitude as the amount of 9,10-phenanthrenequinone initially dissolved in the same portion of polymerization solution. Visual inspection of GR electrode revealed the formation of a stable and well-adhered polymer layer because with the increasing number of potential cycles, the electrode surface becomes darker and less mirror-like. Repeatability of electrochemical polymerization of poly-PQ was analyzed by the evaluation of cyclic voltammograms of seven electrodes, and it was determined that repeatability of polymerization is relatively good, with only an insignificant mismatch after the 18th polymerization cycle.

#### 3.1.2. Investigation of Polymerization by Electrochemical Impedance Spectroscopy

To get a better understanding of electrochemical polymerization of PQ, the electrochemical impedance spectroscopy (EIS) spectra were recorded before and after a selected number of potential cycles. The gathered data of high (40.0 kHz–0.372 Hz) frequency spectra parts are presented in Figure 2. Impedance spectrum was recorded in the frequency range from 40 kHz to 5 mHz at working electrode potential of 0 V vs. Ag/AgCl/KCl_(3M KCl)_ with 10 mV oscillation amplitude. This oscillation potential was selected in order to get conditions similar to that applied during the analysis, which was conducted later in the aqueous solution; moreover, similar EIS parameters as the most optimal were applied in our previous research for the evaluation of other polymer-modified electrodes [37].

From the data presented in Figure 2 it can be observed that all EIS spectra—except that recorded before potential cycling with not modified GR electrode—have a distinctive additional curvature, which becomes more obvious with each cycle. Figure 2 shows the higher frequency parts of the EIS spectra; therefore the data represents electrochemical events happening in a relatively small volume for the ions to respond to the rapidly changing potential so this is most likely happening at close proximity to the surface of the working electrode. The presence of additional feature in the EIS spectra is related to the formation of a new interface between the electrode and the phase of polymerization medium. The increasing curvature of EIS spectra can be explained as a decrease of charge-transfer resistance. Because it is known that the electrode surface is covered by a poly-PQ and the layer of poly-PQ is growing by increasing number of potential cycles, it can be concluded that the polymer is conducting and facilitates the transfer of the charge through phase limit. Apart from additionally curved spectra sequence, the spectrum recorded before the polymerization also provides interesting details: it is seen that in this spectrum the straight line is not tilted by a 45° angle, this means that it is just a part of a second, very large semicircle, thus meaning that the GR electrode surface is involved in charge transfer from ions present in solution. The slight change of the starting points of spectra gathered before and after potential cycling shows only a slight decrease in Ohmic resistance, this fact reveals that during electropolymerization no additional charges have formed in the polymerization solution.

After fitting the gathered impedance data to a selected equivalent circuit (Figure 3) more information on electrochemical polymerization details is extracted (Table 1). Comparing the data for each element characteristics a set of conclusions can be made:

(i) Starting with the Ohmic resistance (R_Ω_) it can be seen that the value does not change much, but tends to slightly decrease with each polymerization cycle, meaning that the ion concentration in the polymerization media is increasing and some additional ions might be generated in the process of electropolymerization. The R2 value is denoting the charge-transfer resistance on the surface to media interface changes very significantly, because the primary value was hundreds of times greater than that registered after 3 cycles and the measurement after 6 CV cycles; these changes most probably are related to increasing electrode surface area, evolved topography and advanced ability to transfer charge via the interface between electrode and solution. When the layer of growing polymer is formed on the electrode surface, then the equivalent circuit is virtually supplemented with additional elements of equivalent circuit, which were monitored as charge-transfer resistance of the GR electrode/polymer interface (R3). From data values it is seen that up to the 9th polymerization cycle a linear increase of resistance-R3 is observed, which is indicating steady polymer layer change, in this case the thickness growth. However, after the 9th cycle the increase of resistance-R3 growth stops, which indicates that the growth of polymer stops, or at least the rate significantly decreases.

(ii) Comparing the capacitance change of the surface interfacing the polymerization solution (CPE2) and interfacing the GR surface (CPE3) it can be seen that CPE2 does not change much and exhibits linear increase from the 3rd to 9th polymerization cycles, meaning that layer shows a quite steady surface and/or adsorbed charge carrier increase. This is most likely due to the linear increase in polymer layer thickness and since it is porous, charges get adsorbed onto the inner surface, within the cavities. While the CPE3 is changing significantly, during initial polymerization cycles it increases, which indicates the initial phase of polymer layer formation on the GR surface; during the next phase, when the thickness of polymer layer increases, the CPE3 decreases significantly. The capacity degree (n2) attributed to CPE2, also is varying during the course of polymerization. A linear decrease of n2 is observed from the 3rd to 18th polymerization cycles and this effect most likely is attributed to the increase in surface roughness, which represents a decrease of ability to accumulate electrical charge at the interface. On the other hand, the capacitance of GR electrode/polymer interface changes more intensively, which indicates complex changes of polymer inner structure during polymerization. At the same time, the capacity degree n3 is linearity decreasing up to the 9th polymerization cycle. The most suitable explanation for this is that due to the increasing formation of complex shape polymer structures the charge movement is restricted or highly impeded from some part of the surface disabling the formation of a highly ordered charged layer.

(iii) The Warburg element values, which are providing information about the diffusion that takes place in the polymerization medium and evaluated by the parameters WR and WT, also demonstrates change in values. The Warburg resistance (WR) linearly decreases, between the 3rd to 9th polymerization cycles due to increase of charge carrier concentration during electrochemical polymerization reaction. Warburg capacitance (WT) also tends to decrease, with maintaining the same linearity from 3rd to 9th polymerization cycles, this is also related to the formation of polymerization products in the polymerization bulk solution.

### 3.2. The Evaluation of Formed Poly-PQ Layer Morphology and Some Electrochemical Properties

#### 3.2.1. The Analysis of Poly-PQ Layer Topography

Topographic properties were studied by atomic force microscopy (AFM) and scanning electron microscopy (SEM). For the poly-PQ layer topography evaluation, two modified electrodes were prepared: first was modified with poly-PQ by one cycle of potential cycling while the second was modified by three cycles. One additional electrode with a bare GR surface was used as a control.

Comparing bare and modified GR electrodes′ SEM images (Figure 4) at magnifications by 50 times (A, B, C), 1000 times (D, E, F), 10,000 times (G, H, I) and 30,000 times (J, K, L) distinctive surface changes are noticed. After initial coating of GR electrode surface by poly-PQ applying 1 potential cycle, visual comparison of modified and bare GR (images A, D, G, J for comparison) electrode reveals that the surface of modified electrode appears to be significantly darker, but differences in SEM images can only be noticed in parts D, E, and F, that are represented by the magnification of 1000 times. While coating the GR electrode surface by three potential cycles the changes of formed poly-PQ layer can be noticed visually and in SEM images at 50 times magnification (parts A, B, C). The main property of poly-PQ-modified electrode surfaces is that polymer layer is composed of poly-PQ flakes with sharp edges, these features by increase of number of polymerization cycles are ‘growing’ first at two dimensions parallel to the surface (E, H, K) and later in all three dimensions (F, I, L).

During the evaluation of modified and bare GR electrode surfaces with AFM the best images were registered using contact mode. Due to technical limitations AFM images were registered only of bare GR electrode surface and after poly-PQ formation-using 1 polymerization cycle. The AFM-based evaluation of poly-PQ-modified electrode surface, which was modified by three polymerization cycles, was not possible because the variations in height were out of measurable scale.

Even though the compared unmodified and modified GR electrode AFM images in Figure 5 are of different size, it can be seen that the height differences of features formed on the electrode surface are disproportionately different. Additionally, in AFM images the roughness of bare GR electrode is well observable (Figure 5A,C) that was almost invisible in SEM images (Figure 4A,D,G,J). In Figure 5B,D (2D and 3D views) it is also seen that poly-PQ layer is composed of sharp flakes similar to that observed in SEM images (Figure 4B), but these flakes are not only parallel to the surface, they also appear to be formed in various orientations. It was also estimated that mean surface roughness (R_q_) of the bare GR electrode is equal to 63.4 nm and the mean surface roughness of the poly-PQ modified electrode is 361 nm.

Summing up the data gathered during the investigation of topography, it can be concluded that the primary clean bare GR electrode surface is not ideally flat and modification by poly-PQ increases the roughness of electrode surface. The GR electrode modification with poly-PQ using cyclic voltamperometry and conditions mentioned earlier in this work, proceeds with 2D flake formation that are orientated not always parallel to the surface after 1st and possibly 2nd polymerization cycle. Later poly-PQ layer growth proceeds with full 3D fractal formation that also tends to be grainier.

#### 3.2.2. Electrochemical Investigation of Poly-PQ Modified Electrode

In this part of investigation cyclic voltamperometry and electrochemical impedance spectroscopy were used for the evaluation of GR electrode modified with poly-PQ. After washing the electrode, it was immersed in an aqueous acetate-phosphate buffer solution (A-PBS), pH 6.0, and electrochemical investigations were performed. Such kind of medium is a most preferred environment for gaining performance of electrochemical sensors and biosensors and other water tolerating systems of higher complexity, where the studied electrodes modified with poly-PQ are most likely to be used.

##### Poly-PQ Modified Electrode Analysis by Cyclic Voltamperometry

Electrochemical conditions were selected in accordance to the conditions mentioned in the literature for the characterization of poly-1,10-phenanthroline monohydrate (PPMH) modified electrodes. Because the mentioned potential interval was from −0.8 V to −0.1 V vs. Ag/AgCl/KCl_(3M KCl)_, to get more insight of the modified electrode electrochemical properties and a better shape of CV cycles the potential sweep interval was expanded symmetrically towards higher positive potentials, making potential interval range from −0.8 V to 0.8 V. During evaluation of electrode modified by poly-PQ applying 9 potential cycles, a perfect CV with clear oxidation and reduction peaks was registered (Figure 6A).

Cyclic voltammograms of poly-PQ modified and bare graphite electrode in A-PBS, pH 6.0, (Figure 6A) illustrates that the formation of a poly-PQ layer on a GR surface greatly enhances its oxidative and reductive properties. Poly-PQ modified GR electrode has a reduction peak at −0.34 V and an oxidation peak at 0.1 V with a relatively high current at both of them, making the poly-PQ a good charge-transferring agent. Additionally, comparing the studied poly-PQ-modified GR electrode with the PPMH modified GR electrode of about the same surface area [32], pH value and scan rate it can be noticed that the electrode studied in this work possesses higher oxidation and reduction peaks, since in the case with PPMH the peaks were of less than 0.1 μA compared to the 0.2 mA measured in this work; in addition to that, the peaks are closer to 0 V potential vs. Ag/AgCl/KCl_(3M KCl)_.

##### Poly-PQ Modified Electrode Analysis by Electrochemical Impedance Spectroscopy

Since the poly-PQ has demonstrated good charge transfer properties it is important to understand its other electrochemical properties, which were evaluated by electrochemical impedance spectroscopy. In these experiments bare GR electrode and poly-PQ-modified GR electrode were examined in A-PBS, pH 6.0. The A-PBS solution was chosen also for achieving a better adsorbed ion layer structure on the poly-PQ surface, which helps to evaluate the surface capacitive properties.

Comparing the EIS spectra of bare GR electrode and poly-PQ modified GR electrode in the high frequency part (Figure 6B) it can be noticed that a layer of poly-PQ makes a large difference, while in the lower frequencies (Figure 6C) the difference is much lower. This effect is related to different origins of the spectra at different frequencies. At the highest frequency information on the Ohmic resistance of media are related to large fractality of formed poly-PQ layer, which provides a large surface for ion adsorption, thus decreasing their concentration in the solution and increasing the resistance of the solution. The most significant fact is the unusual shape of the spectra. This kind of characteristic in EIS can only be possible if the system has a negative resistance element [36] (which means current generation at 0 V potential vs. Ag/AgCl), and it can be related to the measured oxidation peak in cyclic voltammograms that is also very close to 0 V and has significant current value at that potential. There is a possibility that the observed current generation is a measured system artifact but since this was observed with two separate methods this possibility is unlikely. The most suitable equivalent circuit for the GR electrode in A-PBS, pH 6.0, ant it is the same as for the electrode in PQ polymerization medium (Figure 3A), the difference was determined only in element values. For the GR electrode covered with poly-PQ also the same equivalent circuit (Figure 3B), which was used for the evaluation of poly-PQ layer formation, was applied. Due to registration of additional features (Figure 6B) the equivalent circuit should be advanced by a negative resistance element, but due to a better possibility of comparing the results, the circuit was kept the same as for the evaluation of poly-PQ formation (Figure 3B).

Comparing the fitted values of equivalent circuit elements (Table 2) some changes were observed. Firstly, the Ohmic resistances of bare GR electrode and poly-PQ-modified GR electrode are very different. Capacitance of polymer/solution interface (CPE2) increases dramatically, since ionic product generation is of low magnitude and it is related to increasing surface area because the porous polymeric layer grows, which appears to increase by a factor of about a 100. The degree of capacitance (n2) changes very slightly. The charge-transfer resistance at the interface poly-PQ/solution increases significantly. This increase is related to electrochemical properties of the deposited poly-PQ layer and the potential at which the EIS spectrum was recorded (0 V vs. Ag/AgCl/KCl_(3M KCl)_). The charge-transfer resistance at the interface poly-PQ/solution can be attributed to the oxidation of the poly-PQ layer at such electrochemical conditions, but is relatively slow due to the high resistance at the polymer/solution boundary. Warburg element, which is characterizing diffusion, with parameters such as: (i) the resistance (WR) of poly-PQ-modified GR electrode decreases in comparison to that of bare GR electrode and (ii) the capacitance (WT) of poly-PQ-modified GR electrode decreases. Both effects are indicating higher charge mobility.

## 4. Conclusions

During this research it was determined that the optimal electrochemical polymerization conditions for 9,10-phenanthrenequinone on the surface of graphite rod electrode are in the range from 0.5 V to 2.5 V vs. Ag/AgCl/KCl_(3M KCl)_ at a 0.1 V potential sweep rate if polymerization is performed by potential cycling. The poly-PQ forms mainly upon reduction of PQ. EIS results revealed that electrochemical polymerization does not generate a significant amount of new charges (ions) in the polymerization solution. Formed poly-PQ layer very rapidly increases the conductivity of polymer/solution interface by several hundreds of times even after the first three polymerization cycles. From the third to ninth cycles formed poly-PQ layer significantly increases the electrochemically active surface area of electrode and the thickness of poly-PQ layer.

Topographic properties of formed polymer, which were evaluated by AFM and SEM, reveals that formed coating proceeds by the formation of 2D flakes that are orientated not always parallel to the surface at the first and possibly second potential cycle, later poly-PQ layer growth proceeds with full 3D fractal formation that also tend to be grainier. The poly-PQ surface of thickness can be evaluated by SEM, whereas AFM can only be useful for the evaluation of poly-PQ layer formed only by one or two potential cycles. Electrochemical evaluation illustrates that formed poly-PQ layer exhibits very good charge-transfer abilities, because clear oxidation and reduction peaks are observed. It can also be concluded that charge-transfer properties of poly-PQ layer are even better than that of previously investigated PPMH [32]. Summarizing, it can be concluded that poly-PQ is a good charge-transferring agent and the topography of this polymer can be controlled. Therefore, poly-PQ can serve as a potential candidate in the development of future sensors, biosensors and charge storage devices.

## Figures and Tables

**Figure 1 nanomaterials-09-00702-f001:**
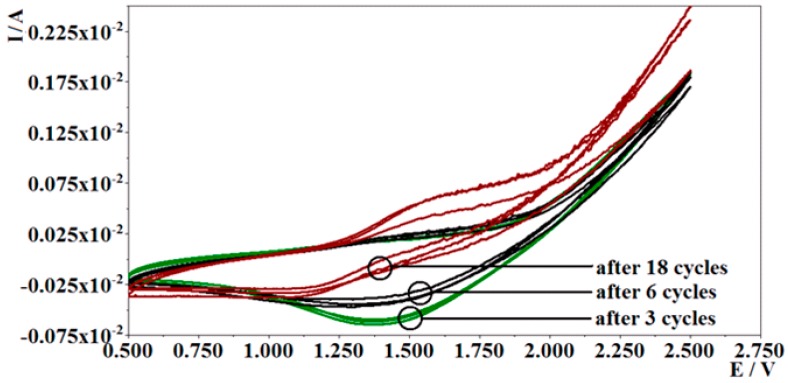
Cyclic voltammograms obtained during potential cycling-based polymerization of 0.001 M PQ in 0.1 M of tetra-*n*-butylammonium perchlorate in acetonitrile, under a scan rate of 100 mV/s vs. Ag/AgCl/KCl_(3M KCl)_.

**Figure 2 nanomaterials-09-00702-f002:**
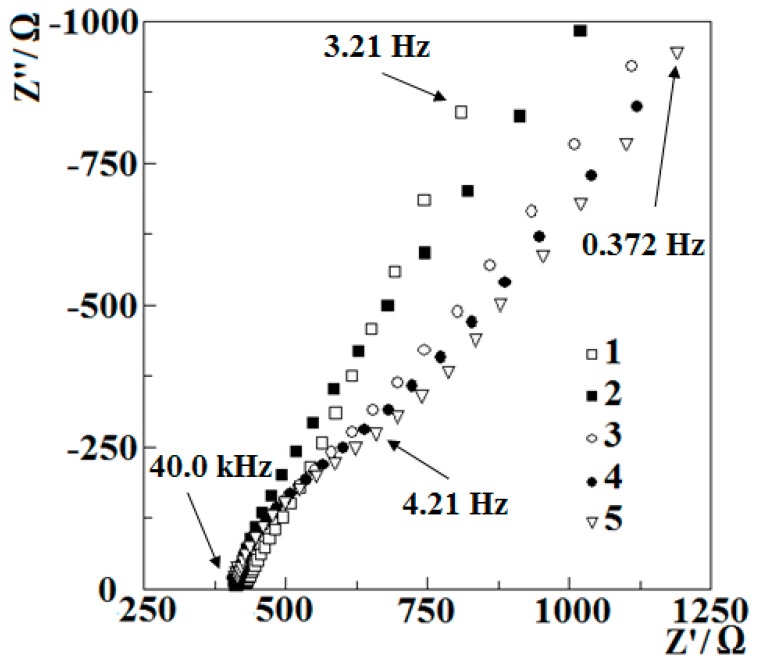
EIS data registered during the polymerization, 1—recorded before polymerization, 2—after 3, 3—after 6, 4—after 9 and 5—after 18 polymerization cycles. EIS data were recorded in a frequency range of 40.0 kHz–372 mHz at 0 V vs. Ag/AgCl/KCl_(3M KCl)_, in acetonitrile-based solution consisting of 0.001 M PQ and 0.1 M of tetra-*n*-butylammonium perchlorate.

**Figure 3 nanomaterials-09-00702-f003:**
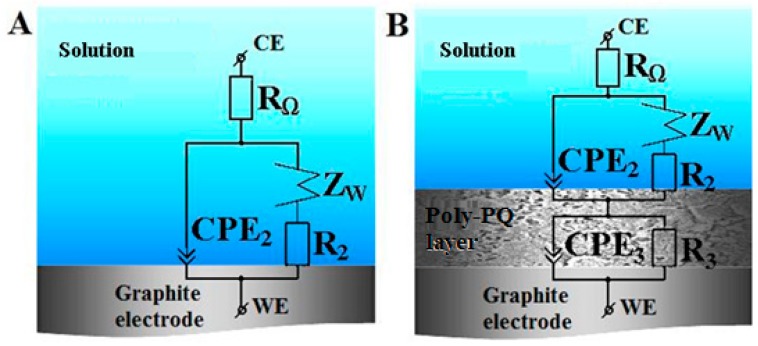
Equivalent circuit for (**A**) clean and (**B**) poly-PQ coated graphite electrode.

**Figure 4 nanomaterials-09-00702-f004:**
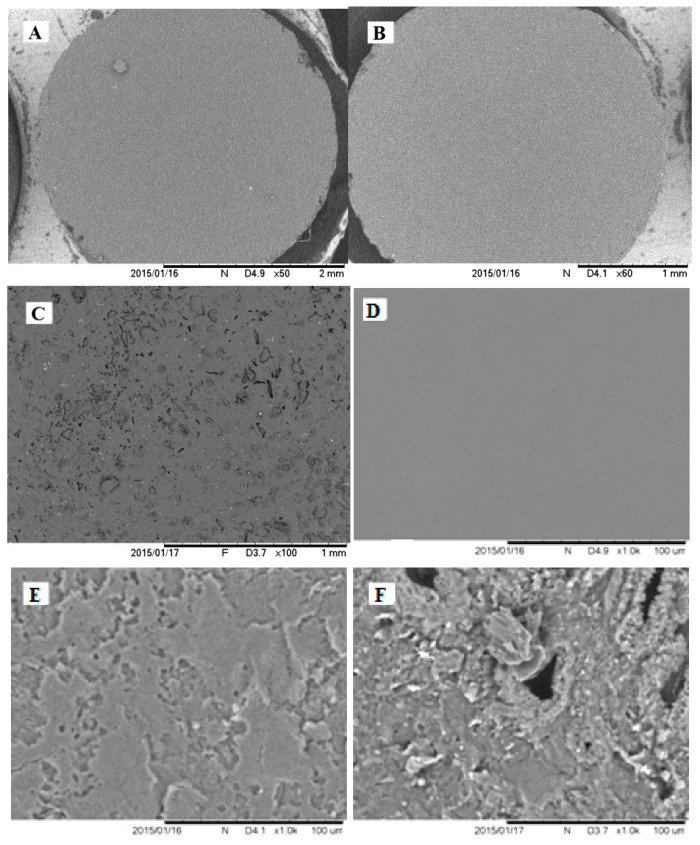
SEM images of GR electrodes: (**A**,**D**,**G**,**J**) bare, (**B**,**E**,**H**,**K**) modified with poly-PQ applying 1 potential cycling, (**C**,**F**,**I**,**L**) modified applying 3 cycles, at magnifications of: (**A**–**C**) 50×, (**D**–**F**) 1000×, (**G**–**I**) 10,000×, (**J**–**L**) 30,000× and scale bar lengths are respectively 1 or 2 mm, 100 μm, 10 μm and 3 μm.

**Figure 5 nanomaterials-09-00702-f005:**
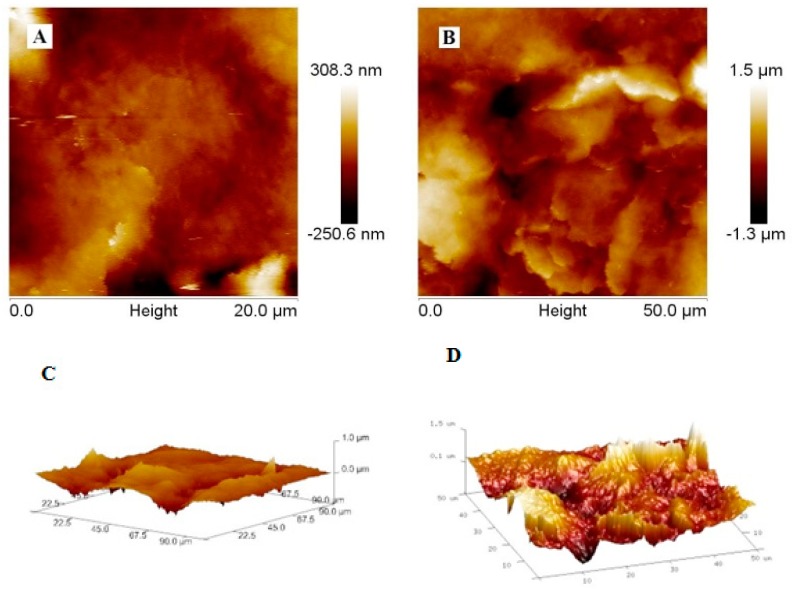
AFM images of GR electrodes: (**A**) bare electrode, (**B**) modified with poly-PQ using 1 polymerization cycle, 2D and respectively (**C**,**D**) 3D projections.

**Figure 6 nanomaterials-09-00702-f006:**
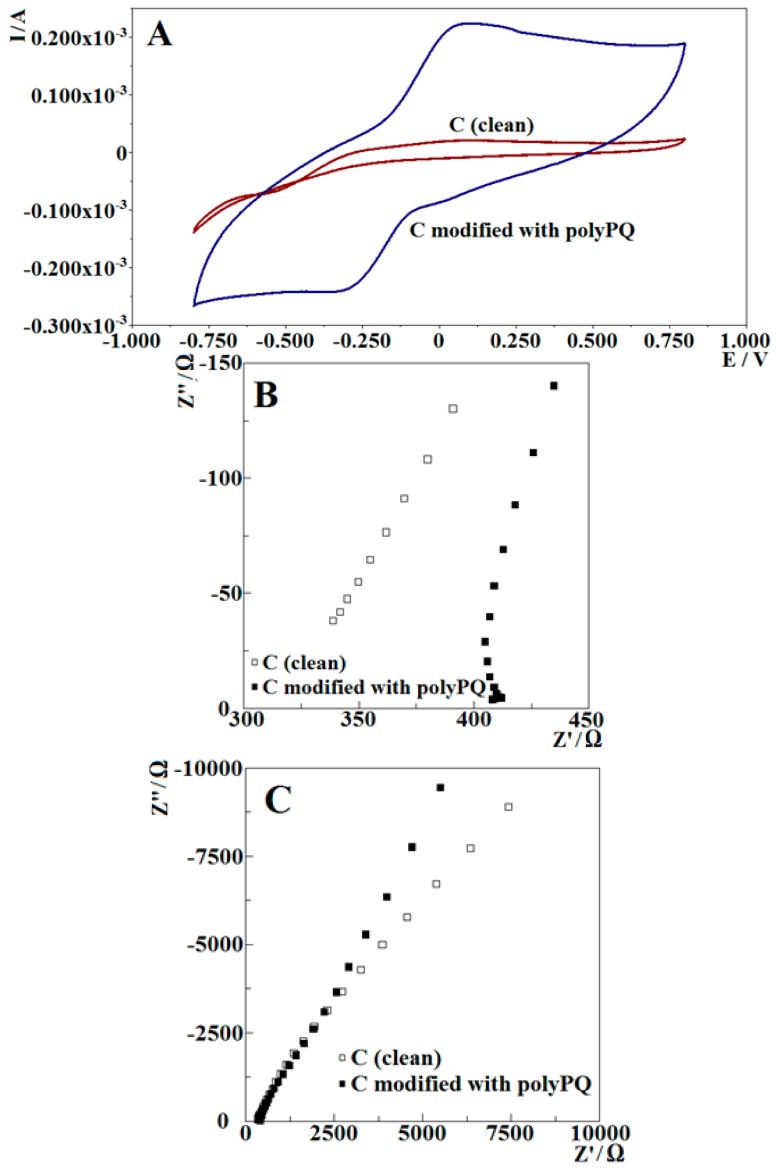
(**A**) Cyclic voltammograms of poly-PQ modified and unmodified GR electrodes at a scan rate of 100 mV/s in pH 6.0 A-PBS; (**B**) EIS spectra of poly-PQ modified and unmodified GR electrodes in high frequencies (40.0 kHz–16.0 Hz) and (**C**) in lower frequencies (16.0 Hz–5.00 mHz) at 0 V vs. Ag/AgCl/KCl_(3M KCl)_.

**Table 1 nanomaterials-09-00702-t001:** Graphite rod electrode under various polymerization steps in polymerization medium.

	R_Ω_, Ω	CPE2, μF	n2	R2, Ω	R3, Ω	CPE3, μF	n3	WR, Ω	WT, F
Before cyclic polymerization	479	126.1	0.7195	281,900				47,860	15
After 3 cycles	414.0	130.0	0.8622	679.8	27.93	1052	0.9857	16,570	11.3
After 6 cycles	412.4	225.0	0.8195	581.7	115.4	1812	0.9672	12,310	10.3
After 9 cycles	411.5	321.8	0.7910	1465.0	207.4	163.6	0.9011	12,320	9.8
After 18 cycles	406.7	404.9	0.6887	713.9	187.3	137.8	0.9500	25,260	10.3

**Table 2 nanomaterials-09-00702-t002:** Equivalent circuit element values of bare and poly-PQ modified graphite electrodes in A-PBS, pH 6.0.

	R_Ω_, Ω	CPE2, μF	n2	R2, Ω	R3, Ω	CPE3, μF	n3	WR, Ω	WT, F
GR (bare)	457	6.203	0.7700	7918				1,475,000	713.5
CR\poly-PQ _(18 cycles)_	385.9	578	0.7294	149,200	530	2068	0.9242	52,470	4.59

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
