# Peer review of "Electrochemical Deposition and Investigation of Poly-9,10-Phenanthrenequinone Layer"

_nanomaterials, 2019, doi:10.3390/nano9050702_

Reviewer 1 Report

The authors report on the electrochemical polymerization of poly-9,10-phenanthrenequinone (PQ) layers on graphite. The paper is overall well written and well organized, and contains a large electrochemical, alectrical and morphological  characterization of the deposited layers.

My main comment is that apparently there is no evidence that the deposited material is actually PQ. The variations of the cyclic voltammograms and of the impedance clearly evidence that something is deposited on the electrode, but no measurements are performed to chemically characterize the layer. A comment on this point would help the readers, not fully expert in electropolymerization, to understand this point.

As minor comments:

-the reduction peak visible in Figure 1 is apparently initially around 1.37 V and not 1.25;

- in page 4, rows 3-5 from the bottom, it seems that the sentence is stopped by a "." and is incomplete. Please check it.

-In page 5 rows 2-5 from the top: the authors comments on the reproducibility of the results but it seems that these measurements are not shown in the manuscript.

Author Response

Response to reviewer #1:

We would like to thank the reviewer for very professional review of our manuscript, valuable comments and recommendations. Thank you for pointing out our mistakes and giving suggestions which further on improve clarity of this paper. We did our best in order to improve the manuscript according to comments and recommendations. All the most important changes are highlighted in the revised manuscript. Corrections and changes are highlighted in the manuscript (in red).

Please   find below short explanations and answers to your questions:

Reviewer #1 wrote:  The authors report on the electrochemical polymerization of poly-9,10-phenanthrenequinone (PQ) layers on graphite. The paper is overall well written and well organized, and contains a large electrochemical, alectrical and morphological  characterization of the deposited layers. My main comment is that apparently there is no evidence that the deposited material is actually PQ. The variations of the cyclic voltammograms and of the impedance clearly evidence that something is deposited on the electrode, but no measurements are performed to chemically characterize the layer. A comment on this point would help the readers, not fully expert in electropolymerization, to understand this point.

Response to Reviewer #1: We will thank the reviewer for highly insightful comments, we did some attempts to discuss formation of PD on the electrode.

Reviewer #1 wrote: -the reduction peak visible in Figure 1 is apparently initially around 1.37 V and not 1.25;

Response to Reviewer #1: Corrected.

Reviewer #1 wrote: - in page 4, rows 3-5 from the bottom, it seems that the sentence is stopped by a "." and is incomplete. Please check it.

Response to Reviewer #1: Corrected.

Reviewer #1 wrote: -In page 5 rows 2-5 from the top: the authors comments on the reproducibility of the results but it seems that these measurements are not shown in the manuscript.

Response to Reviewer #1: Corrected.

We will thank for positive feedback and valuable recommendations.

We hope after all these corrections our manuscript is suitable for publication.

Yours sincerely,
Arunas Ramanavicius

----------------------------------------------------------------
Prof. habil. dr. Arunas Ramanavicius

Head of Department of Physical Chemistry,

Faculty of Chemistry, Vilnius University,

Naugarduko 24, 03225 Vilnius 6, Lithuania; e-mail: [email protected]

Reviewer 2 Report

The paper submitted by A. Ramanavicius et al. describes the electrochemical polymerization on a graphite rod of 9,10 –phenanthrenequinone and its electrochemical and topographic characterization by using different electrochemical and microscopic techniques. The formation of the conductive polymer is performed in acetonitrile by using a potential cycling based method.

The paper is well written and organized, the results being explained in details.

The paper could be accepted in its current form after some minor corrections:

1.      I suggest to add to figure captions the experimental parameters (electrolyte, pH, scan rate, frequency range etc)

2.      In the legend of fig 6A please add the scan rate

3.      An explanation regarding the lack of redox probe in EIS studies will be useful for the journal readers.

Author Response

Response to reviewer #2:

We would like to thank the reviewer for very professional review of our manuscript, valuable comments and recommendations. Thank you for pointing out our mistakes and giving suggestions which further on improve clarity of this paper. We did our best in order to improve the manuscript according to comments and recommendations. All the most important changes are highlighted in the revised manuscript. Corrections and changes are highlighted in the manuscript (in red).

Please  find below short explanations and answers to your questions:

Reviewer #2 wrote:  The paper submitted by A. Ramanavicius et al. describes the electrochemical polymerization on a graphite rod of 9,10 –phenanthrenequinone and its electrochemical and topographic characterization by using different electrochemical and microscopic techniques. The formation of the conductive polymer is performed in acetonitrile by using a potential cycling based method.

The paper is well written and organized, the results being explained in details.

The paper could be accepted in its current form after some minor corrections.

Response to Reviewer #2: We will thank the reviewer for highly insightful comments.

Reviewer #2 wrote: I suggest to add to figure captions the experimental parameters (electrolyte, pH, scan rate, frequency range etc).

Response to Reviewer #2: Corrected.

Reviewer #2 wrote: In the legend of fig 6A please add the scan rate.

Response to Reviewer #2: Corrected.

Reviewer #2 wrote: An explanation regarding the lack of redox probe in EIS studies will be useful for the journal readers.

Response to Reviewer #2: Corrected.

We will thank for positive feedback and valuable recommendations.

We hope after all these corrections our manuscript is suitable for publication.

Yours sincerely,
Arunas Ramanavicius

----------------------------------------------------------------
Prof. habil. dr. Arunas Ramanavicius

Head of Department of Physical Chemistry,

Faculty of Chemistry, Vilnius University,

Naugarduko 24, 03225 Vilnius 6, Lithuania; e-mail: [email protected]

Reviewer 3 Report

A well laid out and readable contribution. It has no great scientific merit but is a useful addition to the body of knowledge on modified electrodes

Author Response

Response to reviewer #3:

We would like to thank the reviewer for very professional review of our manuscript, valuable comments and recommendations. Thank you for pointing out our mistakes and giving suggestions which further on improve clarity of this paper. We did our best in order to improve the manuscript according to comments and recommendations. All the most important changes are highlighted in the revised manuscript. Corrections and changes are highlighted in the manuscript (in red).

Please  find below short explanations and answers to your questions:

Reviewer #3 wrote:  A well laid out and readable contribution. It has no great scientific merit but is a useful addition to the body of knowledge on modified electrodes.

Response to Reviewer #3: We will thank the reviewer for highly insightful comments.

We will thank for positive feedback and valuable recommendations.

We hope after all these corrections our manuscript is suitable for publication.

Yours sincerely,
Arunas Ramanavicius

----------------------------------------------------------------
Prof. habil. dr. Arunas Ramanavicius

Head of Department of Physical Chemistry,

Faculty of Chemistry, Vilnius University,

Naugarduko 24, 03225 Vilnius 6, Lithuania; e-mail: [email protected]